# Direct observation of intrinsic room-temperature ferroelectricity in 2D layered CuCrP$_2$S$_6$

Weng Fu Io[1], Sin -Yi Pang[1], Lok Wing Wong[1], Yuqian Zhao[1], Ran Ding [1], Jianfeng Mao[1,2], Yifei Zhao[1], Feng Guo[1,2], Shuoguo Yuan [1], Jiong Zhao [1], Jiabao Yi [4] & Jianhua Hao [1,2,3] ✉

Multiferroic materials have ignited enormous interest owing to their co-existence of ferroelectricity and ferromagnetism, which hold substantial promise for advanced device applications. However, the size effect, dangling bonds, and interface effect in traditional multiferroics severely hinder their potential in nanoscale device applications. Recent theoretical and experimental studies have evidenced the possibility of realizing two-dimensional (2D) multiferroicity in van der Waals (vdW) layered CuCrP$_2$S$_6$. However, the incorporation of magnetic Cr ions in the ferroelectric framework leads to antiferroelectric and antiferromagnetic orderings, while macroscopic spontaneous polarization is always absent. Herein, we report the direct observation of robust out-of-plane ferroelectricity in 2D vdW CuCrP$_2$S$_6$ at room temperature with a comprehensive investigation. Modification of the ferroelectric polarization states in 2D CuCrP$_2$S$_6$ nanoflakes is experimentally demonstrated. Moreover, external electric field-induced polarization switching and hysteresis loops are obtained in CuCrP$_2$S$_6$ down to ~2.6 nm (4 layers). By using atomically resolved scanning transmission electron microscopy, we unveil the origin of the emerged room-temperature ferroelectricity in 2D CuCrP$_2$S$_6$. Our work can facilitate the development of multifunctional nanodevices and provide important insights into the nature of ferroelectric ordering of this 2D vdW material.

Smart materials possessing diverse fascinating functional properties including ionic conductivity, ferroelectric polarization, and magnetic polarization, are of technological importance for data storage and synaptic device applications[1,2]. The co-existence of ferroelectric and ferromagnetic features in multiferroic materials is even more demanding and promising for both fundamental research and various technical applications[3,4]. Nevertheless, hosting ferroelectric and ferromagnetic orderings simultaneously within a single-phase material is rather challenging, making multiferroic materials relatively scarce. Moreover, conventional multiferroic materials, such as BiFeO$_3$, suffer from the size effect, dangling bonds, and interface effect which make it rather difficult to meet the development demand for device miniaturization with satisfactory device performance, limiting their potential applications[5].

The emerging two-dimensional (2D) van der Waals (vdW) materials, possessing a variety of unprecedented physical properties,

[1]Department of Applied Physics, The Hong Kong Polytechnic University, Hong Kong, P.R. China. [2]The Hong Kong Polytechnic University Shenzhen Research Institute, 518057 Shenzhen, P.R. China. [3]Photonics Research Institute, The Hong Kong Polytechnic University, Hong Kong, P.R. China. [4]Global Innovative Centre for Advanced Nanomaterials, College of Engineering, Science and Environment, The University of Newcastle, Callaghan, NSW 2308, Australia. ✉e-mail: jh.hao@polyu.edu.hk

dangling bond-free surfaces, layered structures with weak interlayer bonding, and the possibility to construct different heterostructures, have shown significant promise for applications in next-generation nanoscale advanced devices[6-8]. This has provoked a surge of research interest to identify and investigate novel 2D materials with multifunctional characteristics for device fabrication. In the past years, several studies have reported the discovery of ferroelastic[9], ferroelectric[10-13], ferromagnetic[14,15], and coexisting ferroelastic–ferroelectric orderings[16,17] in different 2D systems. Such results suggest a possibility of achieving low-dimensional multiferroicity, especially the coexistence of ferroelectric and ferromagnetic orders which is highly attractive from both underlying principles and practical application perspectives. Hence, significant efforts have been involved to explore new possible 2D multiferroic materials and multiferroic has been predicted in several vdW layered materials at the atomic thickness through theoretical calculations[18,19].

Lately, the family of metal thiophosphate (MTP) materials has been receiving special attention owing to their capability to provide a stable platform for incorporating different functional transition metal ions into the structure and exploiting the desired physical properties of materials for various applications[20-22]. The member of the MTP family generally has the chemical formula of $M_1M_2P_2X_6$ where $M_1$and $M_2$ refers to two transitional metals (Cu, Ag, In, Cr, Bi, etc.) and X is a chalcogen (S, Se, etc.). In particular, $CuInP_2S_6$ (CIPS) is the first studied member of the 2D vdW layered MTP family. In CIPS, robust room-temperature ferroelectricity and Cu-ion migration effect have been affirmed[23,24]. Meanwhile, as both the Cu and In cations do not have a partially filled $d$ orbital, no ferromagnetic ordering is expected to present in layered CIPS. Inspiringly, incorporating magnetic Cr cations into the MTP system makes $CuCrP_2S_6$ (CCPS) a potential candidate for realizing low-dimensional multiferroics. Previous theoretical studies showed that the ferromagnetic state of monolayer CCPS is always more stable than the antiferromagnetic and paramagnetic states, implying the possible existence of ferromagnetic order in 2D CCPS[25]. The co-existence of antiferromagnetic and antiferroelectric orderings below the Néel temperature (32 K) and in-plane electrical and magnetic anisotropies were found in CCPS[26-29]. Cajipea et al.[30] examined the structure of CCPS crystal and Cu ion orderings at different temperatures using differential scanning calorimetry and neutron powder diffraction techniques. They indicated that there exist several possible occupation sites of the Cu cations such as the center and off-centered sites within the sulfur octahedral, as well as sites into the interlayer spacing, depending on the temperature. At $T > 190$ K, the CCPS is claimed to own a nonpolar paraelectric structure where the Cu ions are distributed evenly between the up and down sites within the sulfur framework. Lai and co-workers[31] observed a signature of weak ferroelectric behaviors in CCPS flakes at room temperature and they suggest that the polarization may have originated from the electrical-driven Cu ions to the same side of the CCPS layers. However, there lacks strong evidence of proving the ferroelectric characteristics at room temperature and the origin of such emerged ferroelectricity is yet to be explored. The previous experimental results of 2D MTP motivated us to explore whether robust ferroelectricity may exist in ultrathin CCPS layers. Investigating the nature of the ferroelectric ordering in intrinsic vdW layered CCPS is of vital importance for understanding the physical principles as well as probing new possibilities for utilizing the multiple functionalities of CCPS in various devices operating at room temperature.

In this work, we will provide a systematic analysis of evidencing the room-temperature ferroelectric ordering in the intrinsic vdW layered CCPS from both the macroscopic and microscopic aspects. Importantly, we observed strong ferroelectricity along the vertical direction in ultrathin CCPS flakes at room temperature. Through polarization-electric field (*P-E*) hysteresis test and piezoresponse force microscopy (PFM) techniques, a macroscopic spontaneous

polarization of $16.05\,\mu C\,cm^{-2}$ is obtained, and polarization switching behaviors are identified in 2D CCPS down to quadruple layer thickness. Furthermore, we reveal that the ferroelectricity in CCPS originates initially from the off-center vertical arrangement of the Cu cations to the same side within the CCPS layers, resulting in spontaneous electric polarization that can be maintained under ambient conditions down to a few nanometers. Nanoscale devices are fabricated to demonstrate the potential applications of ferroelectric CCPS layers in non-volatile memory storage and the feasibility of constructing heterostructures for multifunctional device applications.

## Results

### Crystal structure of CCPS

Raman spectroscopy, X-ray diffraction (XRD), and transmission electron microscopy (TEM) techniques were performed to confirm the quality and study the crystal structure of ultrathin CCPS. Figure 1a shows the top-view atomic structure of the CCPS crystal. The vdW layered CCPS is consisted of hexagonal sulfur frameworks in which the centers of the octahedral are occupied by Cu, Cr cations, and P–P pairs, forming periodic triangular patterns. Figure 1b presents the high-resolution transmission electron microscopy (HRTEM) image of CCPS showing the in-plane lattice structure with uniform and periodic crystal lattices along identical orientations. The selected area electron diffraction (SAED) pattern in Fig. 1c indicates the large-scale high crystallinity of the CCPS single crystal used in this study. The XRD analysis of CCPS single crystal (Fig. 1d) displays two sharp peaks at 14.35° and 28.35° that correspond to the (002) and (004) planes, respectively, implying a high single crystallinity along the *c*-axis. The Raman spectra of CCPS measured at ambient temperature with thicknesses ranging from 22 nm to bulk crystal are illustrated in Fig. 1e. Noted that the thickness of the CCPS samples selected for Raman analysis is above 20 nm because the samples with thicknesses <20 nm show a very low and indistinguishable Raman peak intensity, whereas increasing the laser power and integration time would damage the ultrathin CCPS samples. There are four observable peaks from the Raman spectra of CCPS, corresponding to the various vibration modes of the $[P_2S_6]^{4-}$ anion framework. The position of the Raman peaks of the CCPS samples with different thicknesses show no obvious changes and are in good agreement with the previously reported vibration modes of CCPS[32]. In addition, the Raman spectra of CCPS at room temperature and above the Curie temperature were measured and plotted in Fig. S1 of Supplementary Information. The four vibration modes decrease slightly in frequency as the temperature increases from 298 to 333 K. This behavior is consistent with other typical materials in which the frequency of the Raman peaks reduces when the temperature increases[33].

### Room-temperature ferroelectricity in CCPS

To study the intrinsic ferroelectricity of CCPS, polarization-electric field (*P-E*) hysteresis tests were conducted. The macroscopic spontaneous polarization of a 0.2 μm CCPS was probed quantitively by *P-E* hysteresis measurement and a typical ferroelectric *P-E* hysteresis loop is obtained with moderate remnant polarization and saturation polarization values of about 14.97 and $16.05\,\mu C\,cm^{-2}$, respectively. Details of the measurement and results are shown in Fig. S2 of the Supplementary Information. In comparison, the measured room-temperature spontaneous polarization value of CCPS in this work is higher than that of CIPS in previous reports[23,34]. To further identify the ferroelectric polarization in ultrathin CCPS nanoflakes, vertical PFM measurements were conducted. The vertical PFM amplitude and phase signals reflect the local electromechanical responses and distinguish different out-of-plane (OOP) polarization orientations of a ferroelectric domain, therefore one can directly observe and adequately confirm the existence of ferroelectricity in the nanoscale. For a comprehensive investigation, the AFM surface topography and PFM OOP

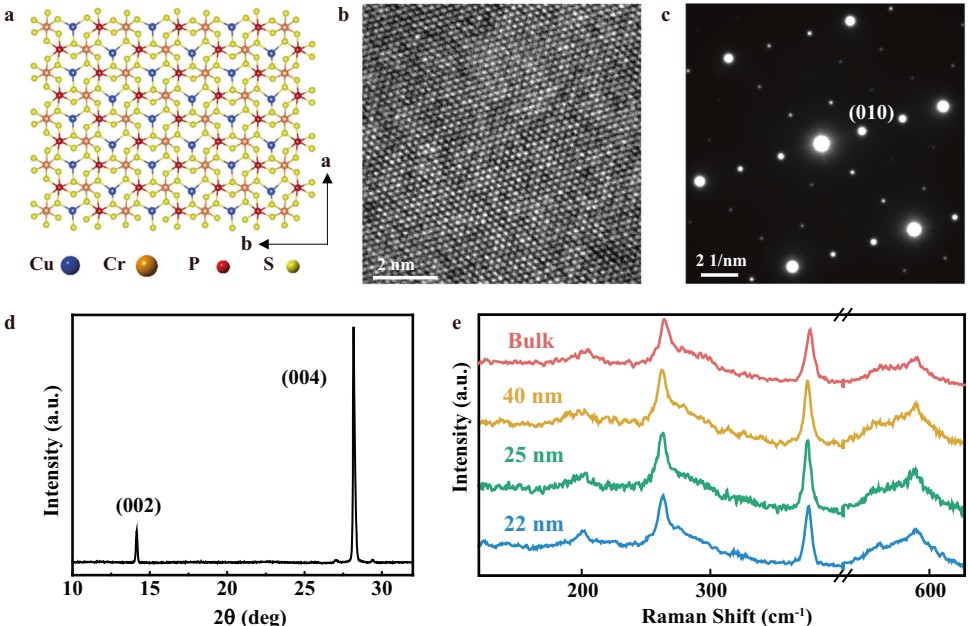

**Fig. 1 | Characterizations of vdW layered CCPS. a** Simulated atomic structure of CCPS from the top view. **b** HRTEM image and **c** SAED pattern of exfoliated CCPS nanoflake. **d** XRD analysis of CCPS single crystal. **e** Raman spectra of exfoliated CCPS samples with different thicknesses from 22 nm to bulk.

piezoresponses of exfoliated CCPS nanoflakes with various thicknesses on conductive Pt substrates are recorded. As shown in the PFM phase image in Fig. S4g of Supplementary Information, the CCPS nanoflakes display the same polarization direction with different thicknesses. While in the other two randomly selected CCPS samples (Fig. S4h–i of Supplementary Information), two distinct areas with ~180° phase difference can be observed across regions with different thicknesses from the PFM phase images, corresponding to the two vertical and oppositely polarized domains. It is suggested from these observations that opposite polarization regions can be formed when CCPS nanoflakes are vertically stacked and the existence of ferroelectricity in ultrathin CCPS is preliminarily verified. On the other hand, it is by definition that ferroelectric materials possess the feature of a switchable polarization direction in response to an external electric field. Thereby, local PFM switching spectroscopy measurements were carried out using conductive PFM tips to affirm the switchability of electric polarization by external bias. Figure 2a and c display the AFM topographic image and the corresponding height profile (along the red line) of ultrathin CCPS nanoflakes down to ~2.6 nm (4 atomic layers). The PFM phase image in Fig. 2b shows a clear phase contrast between the ultrathin CCPS and Pt substrate. PFM off-field hysteresis loops of the quadruple-layer CCPS nanoflake are measured (Fig. 2d), demonstrating typical ferroelectric responses of a butterfly-shaped amplitude loop and phase switching of ~180°. More OOP PFM off-field phase (blue) and amplitude (black) responses of CCPS samples versus applied voltage curves, in which the sample thicknesses range from several to a few hundred nanometers, are shown in Fig. S5 of Supplementary Information. Distinct 180° phase difference and asymmetric butterfly-shaped hysteresis loops are always obtained in the off-field PFM phase and amplitude responses, respectively. These results indicate that OOP ferroelectricity in CCPS can sustain down to a few-layer thickness. The coercive field $E_C$ for polarization switching is extracted from the PFM off-field hysteresis loops of the CCPS samples and plotted against the film thickness in Fig. 2e. The coercive field decreases proportionally in a double logarithmic scale as the sample thickness $t$ increases, and the $E_C$ can be fitted by $E_C \propto t^{-0.929}$. The experimental scaling exponent value deviates from the Janovec–Kay–Dunn (JKD) scaling ($E_C \propto t^{-2/3}$). Deviations from the JKD

scaling have been observed in other ferroelectric thin films and some studies reported a more general inverse power-law correlation between $E_C$ and film thickness[35–37]. For instance, the interfacial "dead layers" and depolarization field effect can enhance the coercive field with decreasing thickness[38,39]. Since all PFM measurements in this study were performed in ambient conditions, some factors such as potential parasitic capacitance might cause undesired voltage drops at the conductive tip–CCPS contact interface thus increasing the $E_C$. The ion migration effect in CCPS could also influence the thickness dependency of the coercive field, resulting in a larger exponent value in the coercive field scaling. Detailed explanation remains for future study, which is of great significance for exploring and understanding the physical mechanism of newly discovered 2D ferroelectric materials.

Next, the intrinsic ferroelectricity is further verified by modifying the ferroelectric domain pattern of CCPS nanoflakes. With the application of an external electric field through conductive PFM tips, square patterns with a phase contrast of ~180° are successfully written on CCPS nanoflakes of different thicknesses as shown in Fig. 3a–c. The corresponding PFM off-field phase (blue) and amplitude (black) hysteresis loops of these CCPS samples are also measured and illustrated in Fig. 3d–f. Noted that the extraction of Cu ions and their deposition on the sample surface can induce changes in the morphology and may give rise to a small difference in the PFM phase responses[28]. Here, no dominant damage or surface protrusion was observed in the topography of the nanoflakes after the domain modification test (Fig. S6 of Supplementary Information), and the samples during characterization were all grounded, excluding possible contributions from other non-ferroelectric artifacts such as ion migration and surface charges. Moreover, the changes in the ferroelectric domain of CCPS nanoflakes under different temperatures were directly monitored using PFM. For the temperature-varying ferroelectric domain investigation, a typical 2D CCPS nanoflake was selected and heated from room temperature up to 338 K. As demonstrated in Fig. 4a, the contrast in the PFM phase slightly decreases when the CCPS nanoflake was heated to 318 K. When the temperature reaches 333 K and above, which is higher than the Curie temperature of CCPS, the ferroelectric domain completely disappears. When the temperature is lowered below the Curie

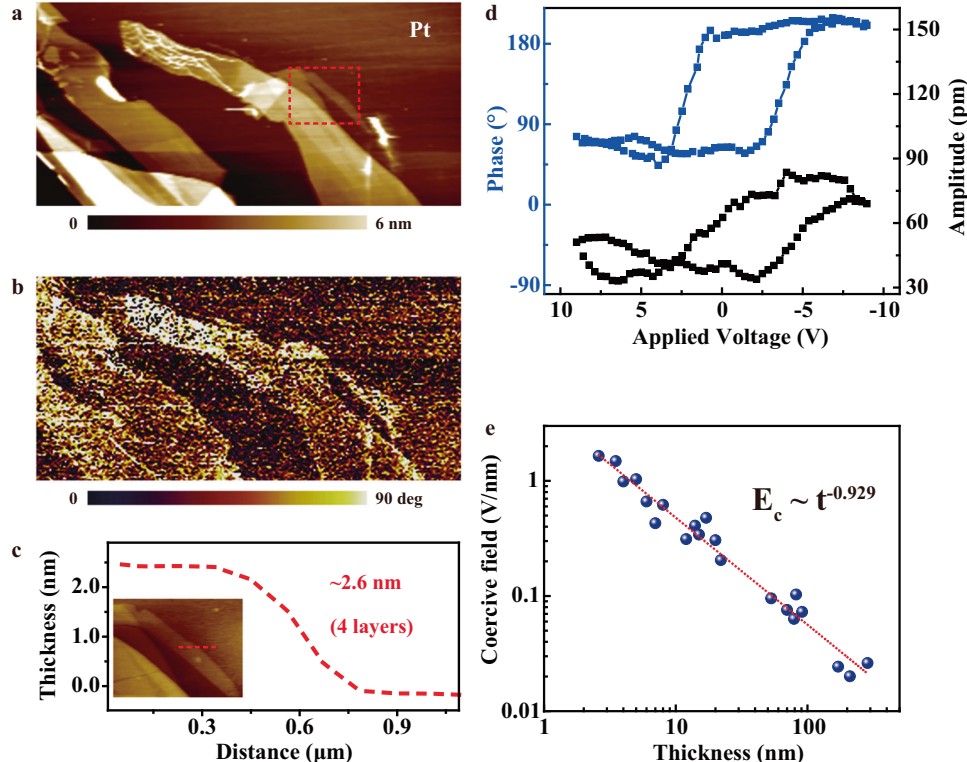

**Fig. 2 | PFM characterizations of ultrathin CCPS nanoflakes. a** AFM topography and **b** corresponding PFM phase response of ultrathin CCPS samples. **c** AFM height profile of a CCPS nanoflake with a thickness of -2.6 nm, corresponds to four CCPS layers. **d** PFM off-field phase and amplitude hysteresis loops of the 2.6-nm-thick CCPS nanoflake. **e** Thickness dependence of the ferroelectric coercive field in double logarithmic scale.

temperature ($T$ < 333 K), the ferroelectric domain reappears. The experimental observations of ferroelectric domain variations at different temperatures sufficiently evince the existence of a reversible transition between ferroelectric and paraelectric phases in 2D CCPS. Second-harmonic generation (SHG) is an effective and non-destructive technique to examine the non-centrosymmetric structure of the CCPS crystal. As displayed in Fig. 4b, the spectrum demonstrates a sharp SHG peak at the half wavelength of the 900 nm excitation laser, indicating the non-centrosymmetric nature of CCPS for their intrinsic ferroelectric characteristics. The intensity of the SHG signal (Fig. 4c) decreases rapidly when the temperature increases above room temperature, where the intensity at 328 K decreases to less than 10% relative to that at room temperature. This implies a gradual disappearance of spontaneous polarization in CCPS and a possible evolution from ferroelectric to paraelectric phase near the Curie temperature, which is a distinct characteristic of ferroelectric materials[23]. It should be emphasized that all the *P-E* hysteresis loops, PFM and SHG measurements were conducted and the ferroelectric behaviors were identified under ambient conditions. Through both macroscopic and microscopic approaches, spontaneous electric polarization along the vertical direction can always be measured in CCPS from bulk crystal down to few-layer thickness under external bias, demonstrating the stable room-temperature OOP ferroelectricity in single-crystalline vdW CCPS layers.

## Origin of intrinsic ferroelectricity

Earlier studies on CCPS imply that the ferroelectric nature might be mainly associated with the ordering of the Cu ions within the CCPS layers[25,26,31]. However, there is a lack of direct microscopic observation to support the mechanism. Herein, aberration-corrected scanning transmission electron microscopy (STEM) was adopted to investigate the atomic structure of pristine CCPS directly exfoliated from the parent crystal, particularly for identifying the Cu locations within the

CCPS layers and probing the origin of the room-temperature ferroelectric effect. The high-angle annular dark-field (HAADF) imaging technique was mainly employed to image the cross-sectional crystal structure at atomic resolution. The HAADF is highly sensitive to the differences in the atomic number ($Z$) of atoms in a material, hence the Cu atoms with the largest atomic number will appear with the greatest brightness and intensity in the HAADF-STEM image. The experimental HAADF-STEM images in Fig. 5a, b show the two opposite ferroelectric polarizations of CCPS, in which the brightest Cu atoms are all situated at the up (down) sites of each vdW CCPS layer, hence inducing a non-zero electric polarization in the upward (downward) direction. The corresponding intensity profiles along the dashed lines in Fig. 5a, b are displayed in Fig. 5c, d, respectively, suggesting again the Cu atoms (with the highest contrast intensity) are situated at the up/down sides within one CCPS layer. Such a result is in good agreement with our proposed structural model of ferroelectric phase CCPS (Fig. 5g) and is strong evidence for the ferroelectricity arising intrinsically from the different off-center orderings of Cu ions in CCPS. Moreover, we observed the ordering of Cu ions at the paraelectric phase by heating the CCPS sample above the Curie temperature. Figure S9 displays the atomically resolved HAADF-STEM image of CCPS at the paraelectric phase where the Z-contrast intensity indicates a partial occupation of the Cu ions evenly at the up and down sites of the CCPS layer, resulting in a zero net polarization. This is consistent with the previously reported structure of CCPS at the paraelectric phase[30,32]. The average thickness of single-layer CCPS is estimated to be about 6.34 Å, smaller than that of the paraelectric phase CCPS at room temperature as reported in the previous literature[32]. Since CCPS and CIPS are sharing the same $[P_2S_6]^{4-}$ anion frameworks as well as Cu cations, the more well-studied ferroelectric CIPS can be adapted to compare and elucidate the structural features of CCPS. Similar behavior of lattice contraction accompanied by the transition to the ferroelectric phase can also be observed in the CIPS sample as noted in the literature[40], which

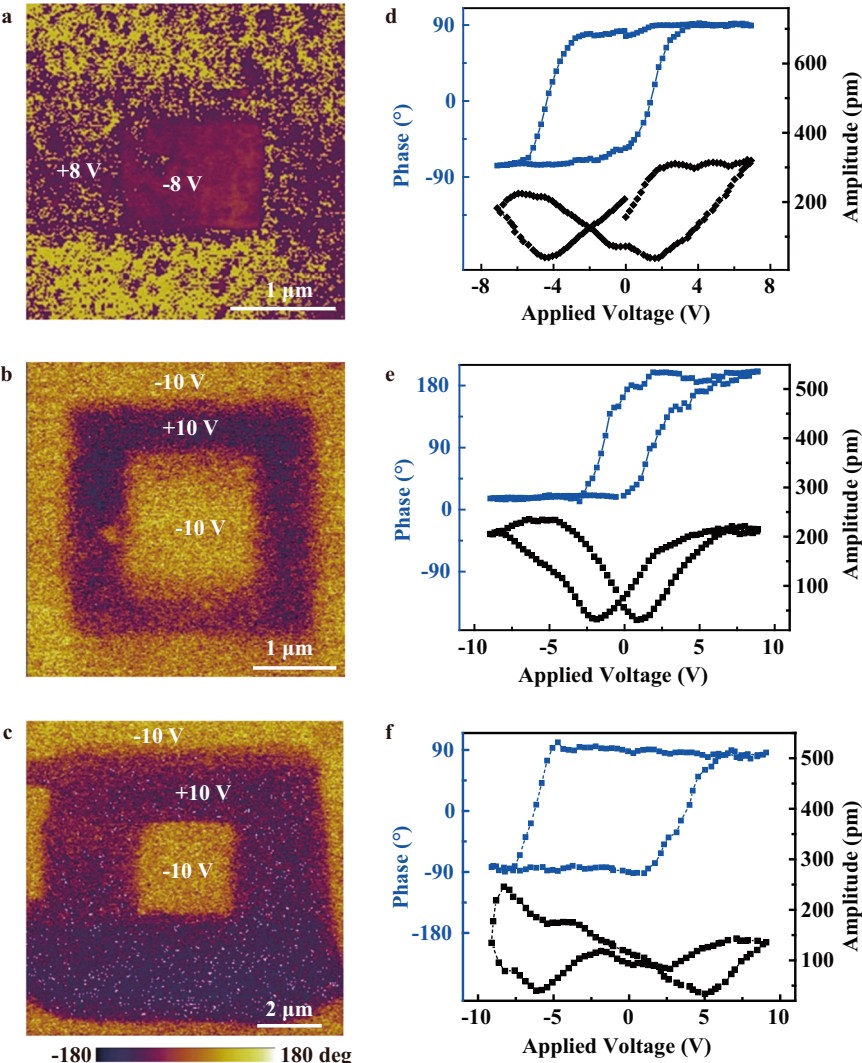

**Fig. 3 | Electric polarization switching in CCPS nanoflakes.** Ferroelectric polarization modification of CCPS nanoflakes with the thickness of **a** 12 nm, **b** 25 nm, and **c** 82 nm, respectively, as imaged by PFM phase mapping by writing box patterns with ±8 and ±10 V. **d–f** The corresponding PFM off-field phase and amplitude hysteresis loops.

correlates to the location of Cu cations on only one side of the ferroelectric CIPS layer and thus reducing the electrostatic repulsion. In comparison, we propose and attribute the difference in the layer spacing between the ferroelectric CCPS in this work and paraelectric CCPS in previous literature to the decrease in electrostatic repulsion as the Cu cations are confined to one side of the CCPS layers. Furthermore, the differential phase contrast (DPC) technique in STEM is also sensitive to the atomic-scale electric field and is capable of visualizing the polarization field in ferroelectric materials[41]. Fig. 5e, f show the colored DPC images of upward- and downward-polarized CCPS samples, while the inset color wheel indicates the direction of the local electric polarization, hence the presence of intrinsic ferroelectricity in CCPS can be directly evinced. A uniform upward electric polarization field (green-yellowish color) can be visualized in Fig. 5e. In comparison, Fig. 5f shows the CCPS sample exfoliated on a Pt substrate in which a polarization field directed from the CCPS layers towards Pt (blue-greenish color) is clearly observed, indicating a downward electric polarization. Overall, the microscopic characterizations indubitably affirmed the ferroelectric phase of CCPS as a stable structure configuration under ambient conditions.

In addition to STEM comparison, density-functional theory (DFT) simulation provided further insight into the charge density difference between the ferroelectric and paraelectric phases of CCPS. Considering the partial Cu occupation as observed by STEM in paraelectric phase CCPS, a representative atomic structure was first selected after comparing the total energies of different possible Cu configurations (details in Supplementary Information). Next, the charge density difference ($\Delta\rho(r)$) between the ferroelectric and the paraelectric phase of CCPS was plotted by subtracting the charge densities: $\Delta\rho(r) = \rho_{\text{ferro}}(r) - \rho_{\text{para}}(r)$, where $\rho_{\text{ferro}}(r)$ and $\rho_{\text{para}}(r)$ are the total charge densities of the ferroelectric and paraelectric CCPS phases, respectively[42]. As shown in Fig. 5h, the yellow/cyan regions correspond to excess/depletion of charges. A net electric dipole along the vertical direction is illustrated, which is in good agreement with the above experimental findings and supports the existence of intrinsic out-of-plane ferroelectricity in vdW layered CCPS.

## Ferroelectric nanodevices

Benefitting from the dangling bond-free surfaces and weak interlayer vdW interactions, 2D materials can easily be integrated into heterostructures and show great promise for various applications. To demonstrate the possible applications of ferroelectric CCPS layers such as nonvolatile memories, ferroelectric tunneling junction (FTJ) consisting of vertically stacked CCPS nanoflake and monolayer

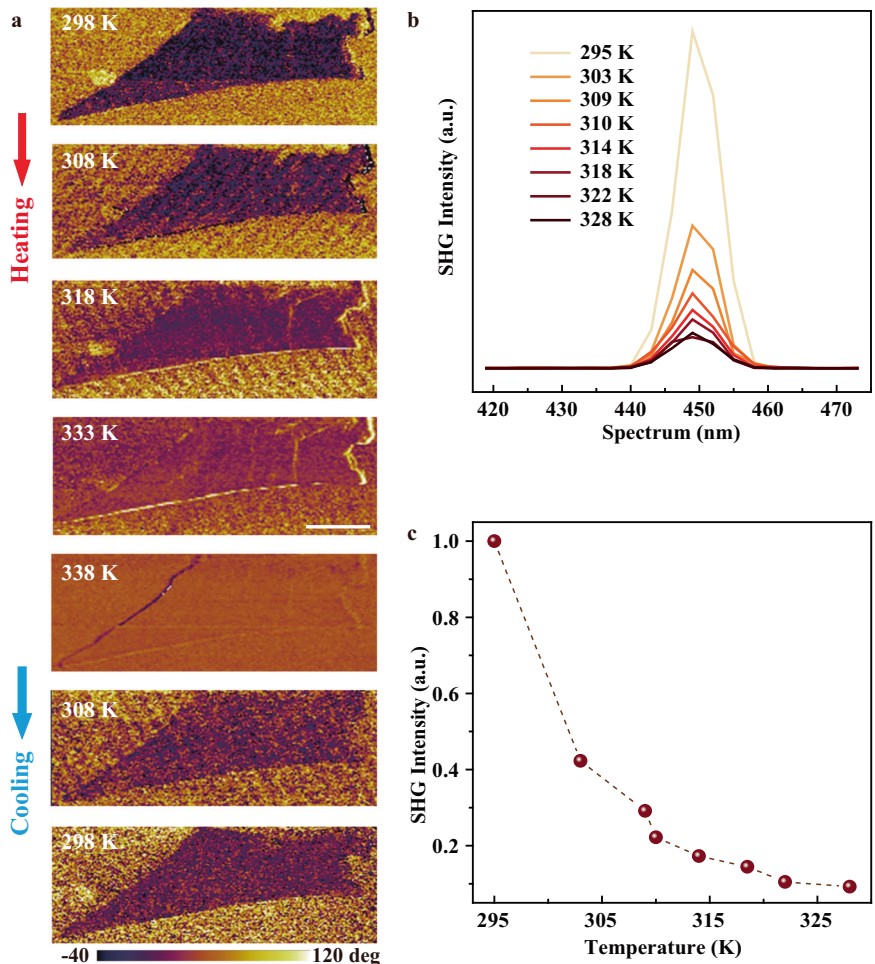

**Fig. 4 | Temperature-dependent characterization of ultrathin CCPS nanoflakes.** **a** Temperature-dependent visualization of the ferroelectric domain in CCPS nanoflake from room temperature to 338 K and then down to room temperature. **b** SHG signals detected from exfoliated CCPS samples at various temperatures and **c** the temperature-dependent SHG peak intensities.

graphene, and vertical ferroelectric diodes based on CCPS nanoflakes and Pt were fabricated. With the use of conductive atomic force microscopy (C-AFM), the electrical transfer characteristics of the CCPS-based FTJ and ferroelectric diode nanodevices were measured. Figure 6a displays the $I$–$V$ curve of a 3.3-nm-thick CCPS/graphene FTJ device after employing an external voltage of ±4 V for polarity switching of the ferroelectric layer. Two distinct resistance states are distinguished, in which the low-resistance state (LRS) and high-resistance state (HRS) represent the ON and OFF states of the non-volatile memory device. The ON/OFF ratio of our 2D CCPS/graphene FTJ device is ~$10^2$, comparable to the performance of the FTJ fabricated based on other conventional ferroelectric materials[43,44]. Besides, we also constructed 2D ferroelectric diodes with ultrathin CCPS layers. The electrical characteristics of a typical ferroelectric diode in Fig. 6b show rectifying behaviors with a similar ON/OFF ratio of ~$10^2$ and the direction of the diode is switchable under external bias. Current rectification behaviors can be observed from devices with different thicknesses of CCPS nanoflakes (Fig. S10 of Supplementary Information), suggesting good repeatability and reliability of the 2D vdW CCPS-based ferroelectric diodes. The endurance of the $I$–$V$ hysteretic switching is examined by repeating the measurement over 20 cycles, and the result is displayed in Fig. 6c. The typical hysteresis behavior and switching voltage of the ferroelectric diode are almost identical, illustrating a good cycle endurance in the $I$–$V$ characteristics of the CCPS-based nanodevices. Figure 6d shows the output current of a

typical CCPS-based ferroelectric diode at the read voltage of 1 V, after writing with +7 V (red) and −7 V (blue). The ON/OFF ratio can still maintain two-order magnitudes after 1500 cycles of reading, suggesting good retention of the ferroelectric polarization state in CCPS. The schematic energy band diagrams of the FTJ and ferroelectric diodes are shown in Fig. 6e, f. The band structure and density of states (DOS) of CCPS were obtained using DFT calculations (details shown in the "Methods" section). According to the calculation results, the band gap of the vdW layered CCPS is about 1.22 eV (Fig. S14 of Supplementary Information). The work function and valence band edge of CCPS nanoflakes are estimated to be about −5.54 and −6.32 eV relative to the vacuum level, respectively, based on our ultraviolet photoelectron spectroscopy (UPS) and scanning Kelvin probe microscopy (SKPM) analyses (details in Supplementary Information). In addition, the exfoliated thin CCPS flake with a thickness of ~280 nm was also fabricated into a vertical diode for illustration of the controllable ferroelectric polarization by an external electric field in bulk counterparts. The electrical transport curve of the ferroelectric diode as shown in Fig. S11 of Supplementary Information also exhibits a characteristic resistance switching phenomenon with a higher switching ratio of ~$10^5$. These results, though acquired with simple and unoptimized device structures, can serve as a proof-of-concept demonstration for the potential application of intrinsic ferroelectric order in 2D CCPS in non-volatile memory devices operating at room temperature.

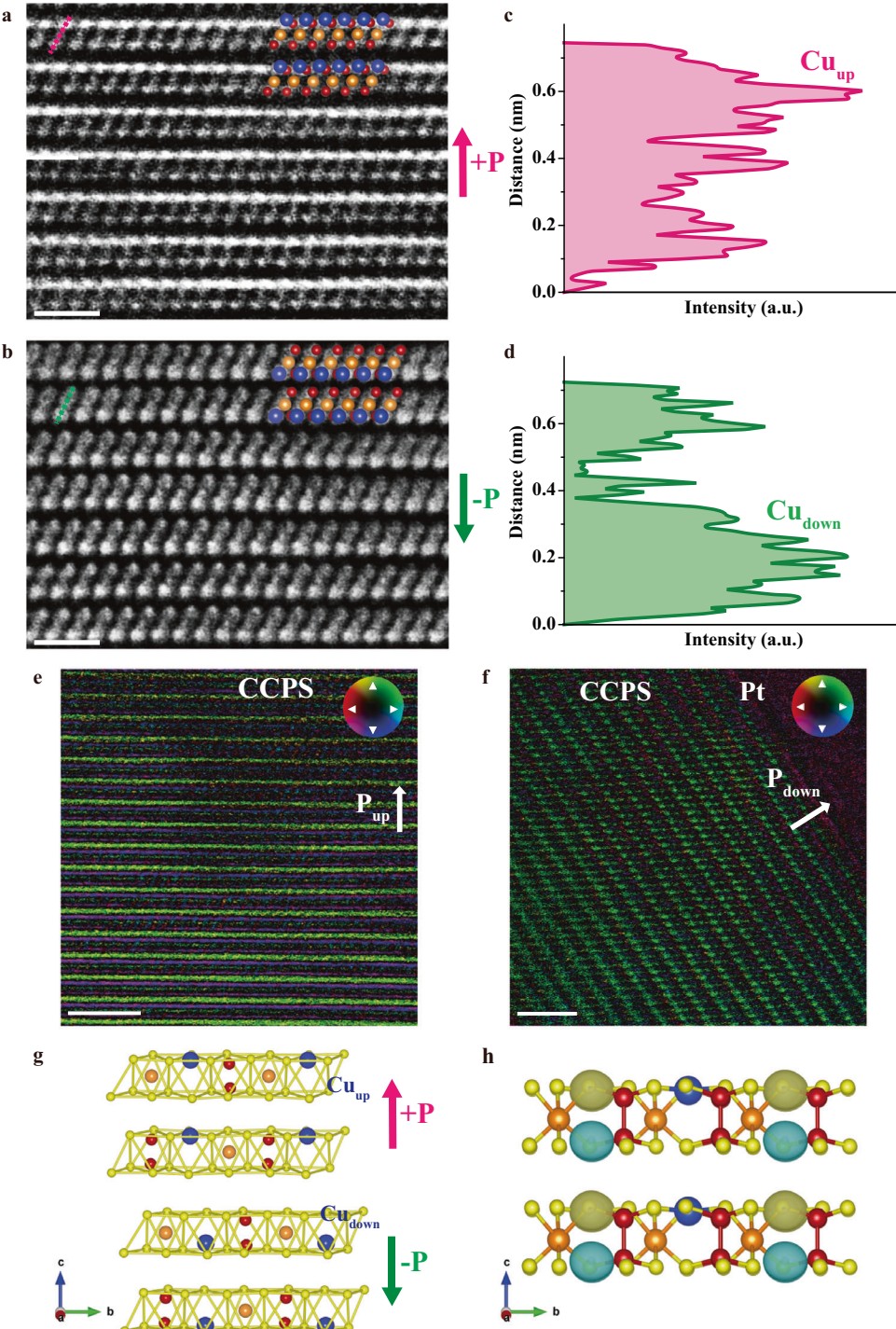

**Fig. 5 | STEM analysis of the cross-section surface of CCPS at the atomic scale.** Atomic resolution HAADF-STEM images of the cross-section of CCPS with **a** positive and **b** negative ferroelectric polarization, respectively. Scale: 1 nm. For simplicity and better comparison, the lightest S atoms are omitted from the simulated structures overlaid on the STEM results. The blue, orange, and red spheres represent Cu, Cr, and P atoms, respectively. **c** and **d** Z-contrast Intensity profiles along the dashed lines corresponding to **a** and **b**, respectively. Colored DPC-STEM images showing uniform **e** upward and **f** downward electric polarization fields in CCPS. Scale: 2 nm. **g** Crystal structure model of CCPS with OOP ferroelectricity. **h** Charge density difference between the ferroelectric and the paraelectric phase CCPS, the yellow/cyan regions correspond to excess/depletion of charges, respectively.

In summary, we reveal the robust ferroelectricity in the 2D vdW layered CCPS at room temperature and explore the origin of the ferroelectric behaviors from an atomic perspective. The spontaneous polarization of CCPS is quantitatively determined to be about $16.05 \, \mu C \, cm^{-2}$, and switching off the electric polarization is achieved in CCPS nanoflakes with the thinnest thickness down to 2.6 nm. Moreover, we discover that the out-of-plane ferroelectricity is intrinsically arising from the off-center ordering of the Cu ions to the same side within the vertical layer, inducing non-zero spontaneous polarization controllable by an external electric field. Our findings shed light on the fundamental nature of the room-temperature ferroelectricity in vdW layered CCPS and enable new opportunities for realizing multifunctional nanodevices.

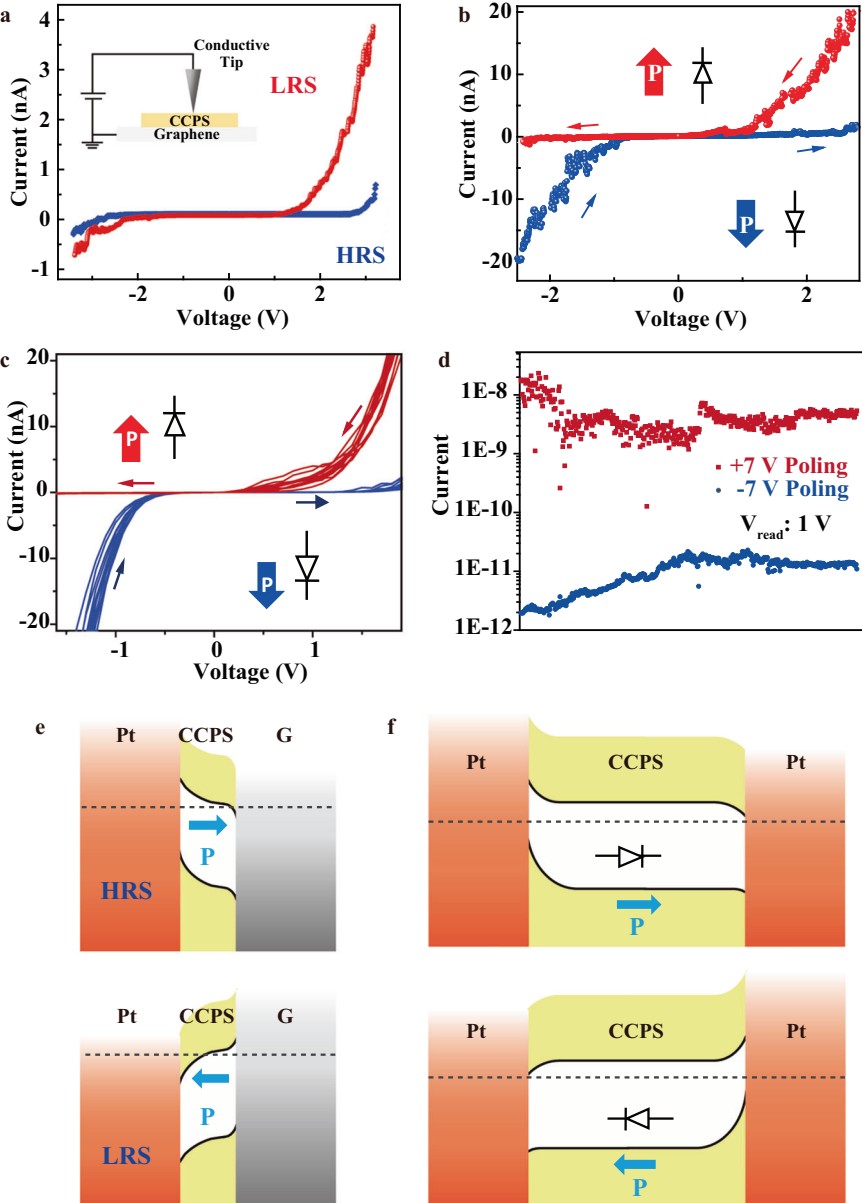

**Fig. 6 | Demonstration of CCPS-based ferroelectric nanodevices. a** *I–V* characterization of a Pt/CCPS/graphene FTJ device with 3.3-nm-thick CCPS by sweeping the voltage bias back and forth. **b** *I–V* curve of a typical Pt/CCPS/Pt switchable ferroelectric diode, in which the thickness of the CCPS nanoflake is about 8.5 nm. **c** Cycling characterization of the *I–V* characteristics and **d** retention measurement of the HRS and LRS states of the CCPS-based ferroelectric diode. Schematics of the band diagram of the **e** FTJ and **f** ferroelectric diode devices.

## Methods

### Sample preparation and characterization of CCPS

Bulk CCPS single crystal was purchased from 2D semiconductors USA (flux growth, highest grade). Ultrathin CCPS samples were prepared through mechanical exfoliation onto arbitrary substrates. The Raman spectra were conducted using the Witec Confocal Raman system with an excitation laser of 532 nm wavelength and 1 μm spot size. Powder XRD analysis was performed using Rigaku smart lab 9 kW (Rigaku, Japan) linked to a 2D detector with Cu K radiation ($\lambda$ = 0.154 nm). UPS analysis was investigated by Nexsa G2 Surface Analysis System (Thermo Scientific Nexsa) with an aluminum Kα X-ray. HRTEM image and SAED pattern were obtained by Orius SC1000 charge-coupled device (CCD) in JEOL JEM-2100F with an accelerating voltage of 200 kV. The atomic-resolution STEM images were acquired by aberration-corrected (S)TEM (Thermo Fisher Spectra 300) with an accelerating voltage of 300 kV. The convergence semi-angle was

14 mrad and the collection angle of the segmented detectors was from 5 to 21 mrad. The electron probe was optimized by a spherical aberration and DCOR corrector via a standard gold sample before observations. The dwell time is 8 μs and the probe current is 16 pA. TEM sample was prepared through drop-casting liquid-exfoliated CCPS ultrathin flakes onto a carbon TEM grid. The cross-sectional STEM sample was fabricated by the focused ion beam technique. The SHG spectra were measured by the Leica TCS SP8 MP confocal microscopy with an excitation wavelength of 900 nm.

### Ferroelectric measurements

The *P-E* hysteresis loops of the exfoliated CCPS nanoflakes were conducted through standard positive-up negative-down (PUND) approach using a probe station (Lakeshore) equipped with a semiconductor parameter analyzer (Keithley 4200A-SCS). The two-terminal vertical Au/CCPS/Au devices for *P-E* hysteresis loop measurements were

prepared by dry-transfer technique. The surface topography, sample thickness, surface potential, piezoelectric and ferroelectric responses were conducted using the different modes of the scanning probe microscope (Asylum MFP-3D Infinity). Piezoelectric and ferroelectric characterizations were examined under dual AC resonance tracking piezoresponse force microscopy (DART-PFM) mode. Pt/Ir-coated conductive tip with a force constant of 2.8 N m$^{-1}$ was used and an AC drive voltage of 1–2 V was employed under a contact resonant frequency of ~340 Hz during all the PFM measurements. Surface potential mapping was conducted using SKPM in alternating AC mode and NAP mode and the delta height is kept at 50 nm from the sample surface.

### Electrical output measurements of devices

The room-temperature electrical transport characteristics of the nanoscale devices were measured under vacuum using a probe station (Lakeshore) with a semiconductor parameter analyzer (Keithley 4200A-SCS) at room temperature.

### Simulation methods

The charge density and electronic properties of CCPS were calculated by CAmbridge Serial Total Energy Package (CASTEP) simulation code with a plane-wave energy cutoff of 700 eV[45]. For the exchange-correlation functional, Perdew–Burke–Ernzerhof (PBE) generalized gradient approximation (GGA) was selected and a set of norm-conserving pseudopotentials was used for the calculation[46]. The CASTEP algorithm was used to optimize the cell, atomic coordinates, and lattice parameters of all MTPs conventional unit cells. The geometry optimization convergence energy tolerance was set to $1.0 \times 10^{-9}$ eV per atom, and the self-consistent field (SCF) for regulating the electronic minimization technique was set to $1.0 \times 10^{-9}$. Moreover, the Broyden–Fletcher–Goldfarb–Shanno (BFGS) method was utilized, with a force and stress convergence tolerance of $0.01$ eV Å$^{-1}$ and $0.02$ GPa for geometry optimization. Additionally, a $12 \times 12 \times 1$ Monkhorst pack $k$-mesh sampling with a separation of $0.0023$ Å$^{-1}$ was employed to sample the Brillouin zone. The simulation results were visualized by VESTA software[47].

### Data availability

The data that support the findings of this study are available from the corresponding authors upon request.

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

## Acknowledgements

This work was supported by the National Natural Science Foundation of China (No. 52233014), HKPFS (ref. no. PF20-46080), Research Grants Council of Hong Kong (PolyU SRFS 2122-5S02), and PolyU Projects of PRI and RCNN (1-CD6X and 1-CE0H).

## Author contributions

W.F.I. and J.H.H. conceived the research work. J.H.H. supervised the research work. W.F.I. characterized the materials and the ferroelectric properties. W.F.I., R.D., and F.G. carried out electrical measurements. L.W.W. conducted (S)TEM analyses. W.F.I. and Y.Q.Z. developed the device fabrication. S.-Y.P. and Y.F.Z. performed the DFT simulations. W.F.I. and J.H.H. wrote the paper. R.D., J.F.M., S.G.Y., J.Z., J.B.Y., and J.H.H. discussed the results and commented on the manuscript.

## Competing interests

The authors declare no competing interests.
