## [Peer Review File · Nature Communications]

Direct Observation of Intrinsic Room-temperature
Ferroelectricity in 2D Layered CuCrP₂S₆REVIEWER COMMENTS

Reviewer #1 (Remarks to the Author):

The manuscript reported the ferroelectricity in the 2D vdW CuCrP2S6 at room temperature. I do not think the ferroelectricity at room temperature is so important, since a lot of materials show ferroelectric polarization at room temperature. In my personal sense, the electric polarization at the thickness of 2.6 nm is interesting, especially in a theoretically predicted multiferroic material. Moreover, the authors show a piece of evidence for the assumption that ferroelectricity is intrinsically arising from the off-center ordering of the Cu ions. The authors also tried to demon the possible device applications. This work can be considered for publication. However, the novelty is not enough and investigation on the critical points is not sufficient. I will give my final decision after a revision.

1. The authors claimed the Cu ordering by the contrast intensity in STEM images. I think the evidence is not enough. More STEM images with different ferroelectric polarization had better to be shown as a comparison. Alternatively, DFT simulated STEM images of different polarization directions are necessary to be compared with experiments. Or any other analysis is necessary.

2. What is the possible reason for the coercive field decreasing as increasing thickness?

3. The discussion on the nanodevice is too schematic. Critical details are missing. For example, what is the sustainability of I-V. Can the I-V loop be same as the first one after some I-V cycles.

4. DFT calculations are not professional. What is the charge density in Fig. 5(f)? Is it a total charge density or conduction band minima (or valence band maxima)? Why is the band structure in Fig. S13 not smooth?

Reviewer #2 (Remarks to the Author):

In this manuscript, the authors reported the intrinsic room-temperature ferroelectricity in CuCuP2S6 (CCPS). The authors utilized P-E loops, PFM, SHG, and STEM to characterize the ferroelectricity in CCPS. In addition, the ferroelectric mechanism is the displacement of Cu atoms in the crystal structure. Furthermore, FTJ and ferroelectric diodes are fabricated to demonstrate the potential applications of ferroelectricity in CCPS. However, some drawbacks hinder its publication.

1 It is well known that CCPS shows antiferroelectricity below $T_c \sim 145$ K (Nat. Commun. 14, 840 (2023)). However, in this manuscript, the authors report the observation of intrinsic room-temperature ferroelectricity in CCPS. P-E loops in this manuscript are lossy capacitor responses, not the typical nonlinear ferroelectric response. Please provide the typical ferroelectric loops, as shown in the references (Nat. Photon. 16, 644–650 (2022), Nat. Mater. 22, 542–552 (2023)).

2 The HRTEM image of CCPS in Figure 1b shows many defects, which may also reduce defect-dipole polarization. Thus, please provide the appropriate data to troubleshoot the impact of the defects.

3 Typically, phase and amplitude data should have a similar inflection point in ferroelectric materials. However, there is a big difference in Figures 2d and 3f. Please explain them.

4 Please provide the corresponding crystal structure model for Figures 5a and b.

5 Based on this manuscript, CCPS is a 2D ferroelectric material. However, other references consider that CCPS belongs to the antiferroelectric phase. Thus, how to realize the controllable synthesis of the ferroelectric and antiferroelectric phases of CCPS? Or does CCPS has the phase transition from antiferroelectric to ferroelectric phase? To demonstrate the intrinsic properties of CCPS, it will be better to provide the corresponding data.

Reviewer #3 (Remarks to the Author):

Although several investigations have been conducted on CuCrP₂S₆ (CCPS), which exhibits several intriguing ferroelectric behaviors compared to other 2D van der Waals materials, such as room-temperature (RT) ferroelectricity (FE) (Y. Lai et al., *Nanoscale* 11, 5163-5170 (2019)) and spin-induced FE based on p-d hybridization (C.B. Park et al., *Adv. Electron. Mater.* 8, 2101072 (2022)), many challenging questions still remain. From an applied physics perspective, it is interesting to investigate whether the ferroelectric properties can be sustained at the atomic layer limit and whether the material can be applied to electronic devices. Furthermore, although RT FE has been observed experimentally and a theoretical model has been proposed (the aligned Cu ions under the electric fields that generate finite electric polarization), the evidence that can support the model is very weak at present status.

In light of these perspectives, the manuscript presents three notable discoveries: (1) the observation of RT FE down to the 4-layer limit, (2) the fabrication of ferroelectric tunneling junctions and diodes with graphene and CCPS, and (3) the presentation of experimental evidence supporting the theoretical model. The findings are impressive, and the experimental data appears to be robust in supporting the results. However, there are several experimental and analytical factors that need to be considered:

1. In order to confirm the spontaneous polarization of both bulk and nanoflake CCPS, the authors conducted polarization-electric field (P-E) loop measurements using the Sawyer-Tower method (L126 - L132 of the manuscript, Fig. S2 and Fig. S3). However, the results (Fig. S2 and Fig. S3) did not show spontaneous, saturated polarization, and the shape of the hysteresis loop was different from that of the phase hysteresis measured by PFM. In my opinion, this could be due to the presence of other factors, especially dielectric or conductive components, which are included in the data along with the pure ferroelectric signal. To avoid this, the authors are encouraged to repeat the measurements using the double-wave method or the 'Positive-Up Negative-Down (PUND)' method to obtain a pure ferroelectric signal.

2. The authors presented the thickness-dependence of the coercive fields (E_c) for polarization switching extracted from the PFM phase-hysteresis loops (L160-L164 of the manuscript and Fig.2(e)). The obtained coercive fields follow a power law fitting, $E_c \sim t^{-a}$ with a value of $a=0.925$. The authors argue that this exponent value is reasonable when compared to other reported ferroelectric films. However, in the references cited (R. Xu et al., *ACS Nano* 12, 4736-4743(2018), R. Nishino et al, *Sci. Rep.* 10, 10864(2020)), the exponents are significantly different ($0.28 < a < 0.8$) from the value obtained for CCPS ($a=0.925$). This casts doubt on the validity of the chosen value for CCPS in the manuscript. Furthermore, the authors did not provide an explanation for why the coercive fields should follow the power law fitting, while such explanations can be found in the cited references. Therefore, the authors should suggest a plausible explanation to justify their choice of

exponent value and provide further analysis of the scaling relation of E_c .

3. To investigate the evolution of FE at high temperatures, the authors conducted second-harmonic generation (SHG) measurements on CCPS nanoflakes, changing the temperature up to 333 K (L185-L194 of the manuscript and Fig. 4). They observed a sharp SHC peak, indicating the non-centrosymmetric nature of CCPS that induces the intrinsic FE, which was suppressed exponentially by increasing the temperature. It is reasonable to accept the explanation that the phase transition from the ferroelectric to paraelectric phase can occur due to the structural modulation, i.e., modification of the sites of Cu atoms by increasing the temperature. However, it is still unclear why the intensity of the SHG peak should decrease exponentially. It would be better if the authors could provide an explanation for this point.

4. The authors utilized HAADF and iDPC-STEM to uncover the origin of RT FE in CCPS and discovered the non-centrosymmetric occupancy of Cu atoms located in the lower side of the atomic layer (L198 - L240 of the manuscript and Fig. 5). These results are in good agreement with the theoretical model suggested by Y. Lai et al. and the SHG results presented in the manuscript. However, to further explain the origin of FE coupled with the SHG results, it would be beneficial if the authors could provide an image showing the change in the location of the Cu atoms at the paraelectric phase.

In addition to the main comments, there are a few minor things that would be better if answered or corrected.

1. In the PFM measurement results on the CCPS samples with different thicknesses (Fig. S5), the difference between the maximum and minimum values of each phase hysteresis is almost 180 degrees, but the values are quite different for each sample. For example, the minimum value of the phase hysteresis in Fig. S5(d) is almost 0 degrees, while that in Fig. S5(h) is almost -90 degrees. Why does this kind of difference happen?

2. It would be better to use the proper unit of polarization ($\mu\text{C}/\text{cm}^2$, not $\mu\text{C}/\text{cm}^{-2}$).

3. Please correct the typos or labeling in the figures, such as omitted labeling in Fig. S4 or typos indicating temperatures in Fig. 4.

REVIEWER COMMENTS

Reviewer #1 (Remarks to the Author):

The manuscript reported the ferroelectricity in the 2D vdW CuCrP_2S_6 at room temperature. I do not think the ferroelectricity at room temperature is so important, since a lot of materials show ferroelectric polarization at room temperature. In my personal sense, the electric polarization at the thickness of 2.6 nm is interesting, especially in a theoretically predicted multiferroic material. Moreover, the authors show a piece of evidence for the assumption that ferroelectricity is intrinsically arising from the off-center ordering of the Cu ions. The authors also tried to demonstrate the possible device applications. This work can be considered for publication. However, the novelty is not enough and investigation on the critical points is not sufficient. I will give my final decision after a revision.

1. The authors claimed the Cu ordering by the contrast intensity in STEM images. I think the evidence is not enough. More STEM images with different ferroelectric polarization had better to be shown as a comparison. Alternatively, DFT simulated STEM images of different polarization directions are necessary to be compared with experiments. Or any other analysis is necessary.

Response: We appreciate for the precious comments from the reviewer. We do agree with the reviewer that more STEM images are needed to support and demonstrate the relation between Cu ion ordering and the ferroelectric origin of CCPS. In response to the issue of insufficient evidence, we have followed the advice from the reviewer. First, additional STEM analysis was performed and HAADF-STEM images of CCPS showing the two opposite ferroelectric polarizations for better comparison were added in the revised manuscript (Figure 5(a) and 5(b)). The intensity contrast of the HAADF-STEM images clearly indicates two diverse orderings of the off-centered Cu ions: one on the upper and one on the lower side of the CCPS layer, corresponding to the up and down ferroelectric polarizations, respectively. This is strong evidence for the origin of ferroelectricity in CCPS that arises intrinsically from the different off-center orderings of the Cu ions.

On the other hand, we have also compared our DFT-simulated ferroelectric CCPS crystal structure with the experimental results from STEM characterization, and the corresponding simulated

crystal structures are overlaid onto the experimental observed HAADF-STEM images in Figure 5 of the revised manuscript. The simulated structures of CCPS at different polarization states match well with our experimental results. We believe the additional atomic-scale STEM analysis can provide sufficient evidence to support the critical points in our work.

Figure R1. Atomic resolution HAADF-STEM images of the cross-section of CCPS with (a) positive and (b) negative ferroelectric polarization, respectively. Scale: 1 nm. The lightest S atoms are omitted from the simulated structures overlaid on the STEM results for simplicity. The blue, orange, and red spheres represent Cu, Cr, and P atoms, respectively.

2. What is the possible reason for the coercive field decreasing as increasing thickness?

Response: We thank the comments from the reviewer. There are several possible factors that can contribute to an increase in the ferroelectric coercive field with decreasing thickness in CCPS. In the past decades, numerous theoretical studies and experimental observations have found the coercive field for the reversal of spontaneous polarization increases with reducing film thickness

in traditional ferroelectric materials.¹⁻⁴ For instance, the Janovec–Kay–Dunn (JKD) scaling model describes the relationship between the coercive field E_C and the film thickness t with $E_C \propto t^{-2/3}$. Based on the experimental values of E_C extracted from PFM hysteresis loops of the CCPS, we plotted the coercive field against film thickness in Figure 2(e) in a double logarithmic scale and obtained a scaling exponent value of 0.929, which deviates from the JKD scaling law. Since all PFM measurements in this study were performed in ambient conditions, some factors such as potential parasitic capacitance might cause undesired voltage drops at the conductive tip-CCPS contact interface thus increasing the E_C . Besides, the effect of substrate clamping and ion migration in CCPS could also influence the thickness dependency of the coercive field. These effects become more prominent as the thickness reduces to the nanoscale, resulting in a larger exponent value in the coercive field scaling. We have added more discussion in the revised manuscript, “The coercive field decreases proportionally in a double logarithmic scale as the sample thickness t increases,some studies reported a more general inverse power-law correlation, resulting in a larger exponent value in the coercive field scaling.” (please see lines 161-172 in pages 4-5 of the manuscript).

3. The discussion on the nanodevice is too schematic. Critical details are missing. For example, what is the sustainability of I-V. Can the I-V loop be same as the first one after some I-V cycles.

Response: Thanks for the insightful comments from the reviewer. Following the suggestion from the reviewer, we have additionally characterized the performance of the CCPS-based ferroelectric nanodevices in more detail, including their sustainability and repeatability. We examined the sustainability of the I - V hysteretic switching by repeating the measurement over 20 cycles, and the results obtained from Pt/CCPS/Pt and Pt/CCPS/graphene ferroelectric diodes are displayed in **Figure R2(a)** and **R2(b)**, respectively. The hysteresis behavior and switching voltage of the ferroelectric diodes are generally consistent and the I - V curves of each cycle are almost the same, showing excellent cycle endurance of I - V characteristics of the CCPS-based nanodevices.

We also studied the retention of the HRS and LRS states of the ferroelectric nanodiode. As shown in **Figure R3**, the ON/OFF ratio can still maintain over two order magnitudes after 1500 cycles of reading, suggesting a good retention ability of the ferroelectric polarization states in CCPS. More detailed discussion on the electrical performance of the nanodevices and corresponding images were added in the revised manuscript, “The endurance of the I - V hysteretic switching is examined

by repeating the measurement over 20 cycles, a good cycle endurance in the I-V characteristics good retention of the ferroelectric polarization state in CCPS.” (please see lines 285-291 in page 7 of the manuscript).

Figure R2. Cycling measurement of the *I-V* characteristics of (a) Pt/CCPS/Pt and (b) Pt/CCPS/graphene ferroelectric diodes.

Figure R3. Retention test of the HRS and LRS states of a typical Pt/CCPS/Pt ferroelectric nanodiode for non-volatile memory applications.

4. DFT calculations are not professional. What is the charge density in Fig. 5(f)? Is it a total charge density or conduction band minima (or valence band maxima)? Why is the band structure in Fig. S13 not smooth?

Response: We thank the helpful comments from the reviewer. It is agreed that the DFT calculations may not be professional enough and lacks refinement. Hence, we have refined our parameters and remodeled the DFT simulations carefully and provided with detailed descriptions and explanations. In response to the charge density difference in Figure 5(f) (rearranged as Figure 5(h) in the modified manuscript), the image is actually a difference charge density diagram which depicts the relocation of charges in CCPS from paraelectric to ferroelectric phase and is plotted by subtracting the charge densities: $\Delta\rho(r) = \rho_{ferro}(r) - \rho_{para}(r)$, where $\rho_{ferro}(r)$ and $\rho_{para}(r)$ are the total charge densities of the ferroelectric and paraelectric phases, respectively. Similar calculation approach has also been employed to visualize ferroelectric polarization.^{5,6} In the updated Figure 5(h), the yellow/cyan regions correspond to excess/depletion of charges, respectively, in which a net electric dipole along the vertical direction is observed, giving rise to the out-of-plane ferroelectric polarization in CCPS.

Figure R4. (a) Calculated band structure and (b) DOS of ferroelectric phase CCPS.

On the other hand, the band structure in Figure S13 has been enhanced by rearranging the Brillouin zone path to maximize the performance and quality of simulations, as described in the previous study.⁷ In response to the relatively rough band structure, the band structure has been recalculated and the mesh of the k -point grid were increased by four times in comparison with our previous calculations. The manuscript and supplementary information have been modified to include the revised images (Figures 5(h) and S14). Accordingly, the corresponding statement and discussion

are added (please see lines 254-261 in pages 6-7 of the manuscript, and page 10 of the supplementary information).

Reviewer #2 (Remarks to the Author):

In this manuscript, the authors reported the intrinsic room-temperature ferroelectricity in CuCrP_2S_6 (CCPS). The authors utilized P - E loops, PFM, SHG, and STEM to characterize the ferroelectricity in CCPS. In addition, the ferroelectric mechanism is the displacement of Cu atoms in the crystal structure. Furthermore, FTJ and ferroelectric diodes are fabricated to demonstrate the potential applications of ferroelectricity in CCPS. However, some drawbacks hinder its publication.

1. It is well known that CCPS shows antiferroelectricity below $T_c \sim 145$ K (*Nat. Commun.* 14, 840 (2023)). However, in this manuscript, the authors report the observation of intrinsic room-temperature ferroelectricity in CCPS. P - E loops in this manuscript are lossy capacitor responses, not the typical nonlinear ferroelectric response. Please provide the typical ferroelectric loops, as shown in the references (*Nat. Photon.* 16, 644–650 (2022), *Nat. Mater.* 22, 542–552 (2023)).

Response: Many thanks for the very valuable comments from the reviewer. It is agreed that the provided P - E hysteresis loops in our original manuscript were unsatisfactory and did not show the typical saturated ferroelectric responses. We have followed the reviewer's suggestion and repeated the P - E hysteresis loop measurements more properly. In order to eliminate the contribution from other non-ferroelectric responses such as dielectric and conductive components, PUND (Positive-Up Negative-Down) approach was adopted instead of the previous Sawyer-Tower method for recording a pure ferroelectric signal. The voltage pulses and current output in PUND measurement are displayed in Figure S2(a) of the revised supplementary information. The calculated P - E hysteresis loop of CCPS was added in the revised supplementary information (Figure S2(b)). Through PUND method, an improved and saturated P - E hysteresis loop response from CCPS sample was obtained which similar to the ferroelectric P - E loops measured in previous references,^{8,9} with remnant polarization and spontaneous polarization of 14.97 and 16.05 $\mu\text{C cm}^{-2}$, respectively.

2. The HRTEM image of CCPS in Figure 1b shows many defects, which may also reduce defect-dipole polarization. Thus, please provide the appropriate data to troubleshoot the impact of the defects.

Response: Thanks for the comments from the reviewer. The HRTEM image is formed by phase contrast. Since the CCPS samples for TEM measurement were transferred onto TEM grids with thermal release tape, many areas of the CCPS sample were not flat, as illustrated in **Figure R5**.

Figure R5. Low-resolution TEM image of the CCPS sample corresponding to the HRTEM image of Figure 1(b) in the original manuscript.

We carelessly made the mistake to provide an HRTEM image in Figure 1(b) taken on an uneven surface of the CCPS sample in which some periodic stripe patterns are induced because of height difference, as displayed on the left side of Figure 1(b). Hence, the HRTEM image of Figure 1(b) did not represent the overall crystal quality of the CCPS used in our study and we have replaced the Figure 1(b) with a HRTEM image measured from a flat CCPS surface. To enhance the reliability of the characterization of the crystal quality, we carefully employed additional HRTEM measurements at randomly selected flat areas of several CCPS samples, and some of the HRTEM results are illustrated in **Figure R6**. The HRTEM images obtained from different CCPS samples always show uniform and periodic lattice structures along the same orientations with negligible defects, indicating the high crystallinity of the CCPS crystal utilized in this work and the possibility of inducing defect dipole polarization in CCPS can be neglected.

Figure R6. (a) Low-resolution TEM and (b) HRTEM images of various CCPS samples.

- Typically, phase and amplitude data should have a similar inflection point in ferroelectric materials. However, there is a big difference in Figures 2d and 3f. Please explain them.

Response: We thank the comment from the reviewer. Figures 2(d) and 3(f) show the PFM phase and amplitude hysteresis loops from different CCPS samples with the thicknesses of 2.6 nm and 82 nm, respectively. The inflection point in the PFM hysteresis loops of Figures 2d and 3f corresponds to the coercive voltage required to switch the ferroelectric polarity of the CCPS layers. When the thickness of the ferroelectric material is reduced to the nanometer scale, the coercive voltage/field usually changes with thickness, while similar thickness-dependent behaviors can also be observed in other traditional perovskite and 2D vdW ferroelectric materials.¹⁰⁻¹² As the polarization axis of CCPS is along the vertical direction, a larger switching voltage does not always mean a larger electric field is applied to the CCPS samples. Therefore, when the PFM hysteresis responses are plotted with applied voltage, the switching points of the phase and amplitude loops of CCPS with different thicknesses show variations.

- Please provide the corresponding crystal structure model for Figures 5a and b.

Response: Thank you for the useful comment from the reviewer. For a better comparison with the experimental results, we have added our DFT-simulated CCPS crystal structure model corresponding to Figure 5 and overlaid onto the experimental observed HAADF-STEM images in

the revised manuscript. In addition, we have also provided HAADF-STEM images illustrating the two opposite ferroelectric polarizations of CCPS in the updated Figure 5(a) and 5(b) of the manuscript for a better comparison.

5. Based on this manuscript, CCPS is a 2D ferroelectric material. However, other references consider that CCPS belongs to the antiferroelectric phase. Thus, how to realize the controllable synthesis of the ferroelectric and antiferroelectric phases of CCPS? Or does CCPS has the phase transition from antiferroelectric to ferroelectric phase? To demonstrate the intrinsic properties of CCPS, it will be better to provide the corresponding data.

Response: Thanks for the comments. First, in response to the realization of controllable synthesis of ferroelectric and antiferroelectric/paraelectric phases, the bulk CCPS single crystal used in our work was directly purchased from the 2D Semiconductors company instead of self-synthesis in our laboratory. The study (*Nat. Commun.* 14, 840 (2023)) mentioned by the reviewer in previous comments mainly focused on the in-plane electrical and magnetic anisotropies in CCPS and did not investigate the room-temperature ferroelectric ordering of CCPS. Both in their work as well as their cited references, the CCPS samples were self-synthesized using the chemical vapor transport (CVT) method. While the CCPS crystal from the 2D Semiconductors company used in our work was synthesized through flux growth with the highest grade. Since the synthesis approaches of CCPS single crystal varied, the phase and crystallinity of the synthesized CCPS crystal might also differ and this might be a possible reason for the ferroelectric phase of CCPS.

On the other hand, the HAADF-STEM images of the CCPS cross-sectional crystal structure displayed in Figure 5 were obtained from intrinsic CCPS samples freshly exfoliated with no pre-electrical poling or other treatments. We have conducted many atomic resolution STEM measurements and did not observe any existence of antiferroelectric structure from the variety of pristine CCPS samples under measurements at room temperature so far. Hence, we believe there is no transition from antiferroelectric to ferroelectric phase at room temperature in the pristine CCPS used in our study.

Reviewer #3 (Remarks to the Author):

Although several investigations have been conducted on CuCrP_2S_6 (CCPS), which exhibits several intriguing ferroelectric behaviors compared to other 2D van der Waals materials, such as room-temperature (RT) ferroelectricity (FE) (Y. Lai et al., *Nanoscale* 11, 5163-5170 (2019)) and spin-induced FE based on *p-d* hybridization (C.B. Park et al., *Adv. Electron. Mater.* 8, 2101072 (2022)), many challenging questions still remain. From an applied physics perspective, it is interesting to investigate whether the ferroelectric properties can be sustained at the atomic layer limit and whether the material can be applied to electronic devices. Furthermore, although RT FE has been observed experimentally and a theoretical model has been proposed (the aligned Cu ions under the electric fields that generate finite electric polarization), the evidence that can support the model is very weak at present status.

In light of these perspectives, the manuscript presents three notable discoveries: (1) the observation of RT FE down to the 4-layer limit, (2) the fabrication of ferroelectric tunneling junctions and diodes with graphene and CCPS, and (3) the presentation of experimental evidence supporting the theoretical model. The findings are impressive, and the experimental data appears to be robust in supporting the results. However, there are several experimental and analytical factors that need to be considered:

1. In order to confirm the spontaneous polarization of both bulk and nanoflake CCPS, the authors conducted polarization-electric field (*P-E*) loop measurements using the Sawyer-Tower method (L126 - L132 of the manuscript, Fig. S2 and Fig. S3). However, the results (Fig. S2 and Fig. S3) did not show spontaneous, saturated polarization, and the shape of the hysteresis loop was different from that of the phase hysteresis measured by PFM. In my opinion, this could be due to the presence of other factors, especially dielectric or conductive components, which are included in the data along with the pure ferroelectric signal. To avoid this, the authors are encouraged to repeat the measurements using the double-wave method or the 'Positive-Up Negative-Down (PUND)' method to obtain a pure ferroelectric signal.

Response: Thanks very much for the very helpful comments from the reviewer. Because of the large leakage current in CCPS thin films, it is difficult to obtain a typical ferroelectric shape in the measured *P-E* loops at room temperature. Very short voltage pulses were used in the Sawyer-

Tower approach which led to unsaturated P - E loops as shown in original Figure S2 and S3. In contrast to the Sawyer-Tower method, the shape of the phase hysteresis loops measured through PFM were spontaneous and saturated as the PFM approach is less affected by leakage current. We have followed the reviewer's suggestion and repeated the measurements of P - E loops from CCPS through a more accurate PUND (Positive-Up Negative-Down) approach for recording a pure ferroelectric signal. The voltage pulses and current output in PUND measurement are displayed in the updated Figure S2(a) of supplementary information. The calculated P - E hysteresis loop of CCPS was added in the revised supplementary information (Figure S2(b)). Through PUND method, an improved and saturated P - E hysteresis loop response from a CCPS sample was obtained which similar to the ferroelectric P - E loops measured in previous references,^{8,9} with remnant polarization and spontaneous polarization of 14.97 and 16.05 $\mu\text{C cm}^{-2}$, respectively.

2. The authors presented the thickness-dependence of the coercive fields (E_c) for polarization switching extracted from the PFM phase-hysteresis loops (L160-L164 of the manuscript and Fig.2(e)). The obtained coercive fields follow a power law fitting, $E_c \sim t^{-a}$ with a value of $a=0.925$. The authors argue that this exponent value is reasonable when compared to other reported ferroelectric films. However, in the references cited (R. Xu et al., *ACS Nano* 12, 4736-4743(2018), R. Nishino et al, *Sci. Rep.* 10, 10864(2020)), the exponents are significantly different ($0.28 < a < 0.8$) from the value obtained for CCPS ($a=0.925$). This casts doubt on the validity of the chosen value for CCPS in the manuscript. Furthermore, the authors did not provide an explanation for why the coercive fields should follow the power law fitting, while such explanations can be found in the cited references. Therefore, the authors should suggest a plausible explanation to justify their choice of exponent value and provide further analysis of the scaling relation of E_c .

Response: Thanks for the comments. The Janovec–Kay–Dunn (JKD) scaling model describes the relationship between the coercive field E_C and the film thickness t , where $E_C \propto t^{-2/3}$. Based on the experimental values of coercive field E_C extracted from PFM hysteresis loops of the CCPS, we plotted the coercive field against film thickness in Figure 2(e) in a double logarithmic scale and obtained a scaling exponent value of 0.929, which deviates from the JKD scaling law. Deviations from the JKD scaling have been observed in other ferroelectric thin films and some studies reported a more general inverse power-law correlation between E_C and film thickness ($E_C \propto t^{-1}$).¹³⁻

¹⁵ For instance, the interfacial “dead layers” and depolarization field effect can enhance the coercive field with decreasing thickness.^{1,16} In this study, the coercive fields were mainly measured from CCPS samples less than 100 nm, while the JKD scaling might not be well-fitted at the nanoscale thickness. Since all PFM measurements in this study were performed in ambient conditions, some factors such as potential parasitic capacitance could cause undesired voltage drops at the conductive tip-CCPS contact interface thus increasing the E_C . The effect of substrate clamping and ion migration in CCPS might also influence the thickness dependency of the coercive field, resulting in a larger exponent value in the coercive field scaling. In response to the insufficient explanation, we have provided additional discussion in the revised manuscript, “The coercive field decreases proportionally in a double logarithmic scale as the sample thickness t increases,some studies reported a more general inverse power-law correlation, resulting in a larger exponent value in the coercive field scaling.” ((please see lines 161-172 in pages 4-5 of the manuscript).

3. To investigate the evolution of FE at high temperatures, the authors conducted second-harmonic generation (SHG) measurements on CCPS nanoflakes, changing the temperature up to 333 K (L185-L194 of the manuscript and Fig. 4). They observed a sharp SHG peak, indicating the non-centrosymmetric nature of CCPS that induces the intrinsic FE, which was suppressed exponentially by increasing the temperature. It is reasonable to accept the explanation that the phase transition from the ferroelectric to paraelectric phase can occur due to the structural modulation, i.e., modification of the sites of Cu atoms by increasing the temperature. However, it is still unclear why the intensity of the SHG peak should decrease exponentially. It would be better if the authors could provide an explanation for this point.

Response: We thank the insightful comments from the reviewer. The SHG is an effective characterization technique for studying both conventional 3D and new 2D vdW ferroelectric materials. In general, thermal fluctuations are generated when temperature increases, destabilizing the spontaneous polarization and driving the material into a high-symmetry structure at a higher temperature. Similar to CCPS, an obvious decreasing trend in the SHG intensity also occurs in other ferroelectric materials when temperature approaches the Curie temperature.^{11,17,18} This suggests a gradual disappearance of spontaneous polarization in ferroelectric materials and a potential occurrence of phase transition to a centrosymmetric structure. In our original manuscript,

we made a careless mistake of inaccurately fitting the SHG intensity curve with an exponential function. In response, we have revised the graph in Figure 4(c) and the associated discussion to improve the correctness of our work, “The intensity of the SHG signal (Figure 4(c)) decreases rapidly when the temperature increases above room temperature, where the intensity at 328 K decreases to less than 10% relative to that at room temperature. This implies a gradual disappearance of spontaneous polarization in CCPS and a possible evolution from ferroelectric to paraelectric phase near the Curie temperature, which is a distinct characteristic of ferroelectric materials.” (please see lines 198-203 in page 5 of the manuscript).

4. The authors utilized HAADF and iDPC-STEM to uncover the origin of RT FE in CCPS and discovered the non-centrosymmetric occupancy of Cu atoms located in the lower side of the atomic layer (L198 - L240 of the manuscript and Fig. 5). These results are in good agreement with the theoretical model suggested by Y. Lai et al. and the SHG results presented in the manuscript. However, to further explain the origin of FE coupled with the SHG results, it would be beneficial if the authors could provide an image showing the change in the location of the Cu atoms at the paraelectric phase.

Response: Thanks for the precious comments from the reviewer. We do agree with the reviewer that the provision of atomic-resolution image showing the Cu ion orderings at the paraelectric phase is more beneficial for understanding the underlying mechanism of the origin of room-temperature ferroelectricity in CCPS. The ferroelectric-paraelectric phase transition is a reversible process depending on temperature. As the STEM at our site is not equipped with *in situ* heating holder, we attempted to *ex situ* heat our CCPS specimen above the Curie temperature and tried to observe the position of Cu ions at the paraelectric phase. The Cu ion ordering of CCPS at the paraelectric phase is shown in the atomic resolution HAADF-STEM image of **Figure R7(a)**. The corresponding intensity profile along the red dashed line (**Figure R7(b)**) shows two highest and identical Z-contrast intensities at the up and down positions, indicating a partial occupation of the Cu ions evenly at the up and down sites of the CCPS layer which results in a zero net polarization. This observation is consistent with the previously reported structure of CCPS at the paraelectric phase.^{19,20}

Figure R7. (a) Atomic resolution HAADF-STEM image of CCPS at the paraelectric phase and (b) the corresponding intensity profile along the red dashed line.

On the other hand, we also observed and compared the location of Cu ions at the two different ferroelectric polarization states using our aberration-corrected STEM equipment, as shown in the HAADF-STEM images of revised Figure 5(a) and 5(b) in the manuscript. The intensity contrast of the HAADF-STEM images clearly indicates two diverse orderings of the off-centered Cu ions: one on the upper and one on the lower side of the CCPS layer, corresponding to the up and down ferroelectric polarizations, respectively. This is strong evidence for the origin of room-temperature ferroelectricity in CCPS that arises intrinsically from the different off-center occupancy of the Cu ions.

In addition to the main comments, there are a few minor things that would be better if answered or corrected.

1. In the PFM measurement results on the CCPS samples with different thicknesses (Fig. S5), the difference between the maximum and minimum values of each phase hysteresis is almost 180 degrees, but the values are quite different for each sample. For example, the minimum value of the phase hysteresis in Fig. S5(d) is almost 0 degrees, while that in Fig. S5(h) is almost -90 degrees. Why does this kind of difference happen?

Response: Thanks for the comments. According to the operation mechanism of the PFM equipment, the difference between the minimum and maximum values of a typical phase hysteresis loop from ferroelectric materials should always be around 180°, which is also obtained in the off-field phase hysteresis responses of all our CCPS samples. The PFM was operated at contact mode

and the measurements were conducted near the contact resonance frequency between the conductive AFM tip and the sample surface (**Figure R8(a)**). The contact resonance frequency might shift by a few kHz on different areas of the CCPS samples during PFM measurement which could cause an off-centering in the phase response as illustrated in **Figure R8(b)**, and similar phenomena were also observed in other conventional and 2D vdW ferroelectric materials.²¹⁻²⁵ Nevertheless, the relative phase position under zero applied voltage shall not affect the characterization of ferroelectricity, and the main consideration should be the difference between the minimum and maximum values of the phase hysteresis loop where a phase shift of $\sim 180^\circ$ is expected from ferroelectric materials.

Figure R8. (a) Tuning of the contact resonance frequency between the conductive AFM tip and the sample surface. (b) Repeated PFM off-field hysteresis loop measurements when the phase is off-centered.

2. It would be better to use the proper unit of polarization ($\mu\text{C}/\text{cm}^2$, not $\mu\text{C}/\text{cm}^{-2}$).

Response: We thank the comment from the reviewer. The proper unit for polarization has been corrected in the revised manuscript.

3. Please correct the typos or labeling in the figures, such as omitted labeling in Fig. S4 or typos indicating temperatures in Fig. 4.

Response: Thanks for the reviewer's careful reading. The typos in Figure 4 were fixed in the revised manuscript. We also added labeling in Figure S1, S4, S6, S13 and S15.

Reference:

- 1 Tagantsev, A. K. & Stolichnov, I. A. Injection-controlled size effect on switching of ferroelectric thin films. *Appl. Phys. Lett.* **74**, 1326-1328 (1999).
- 2 Baudry, L. & Tournier, J. Lattice model for ferroelectric thin film materials including surface effects: investigation on the “depolarizing” field properties. *J. Appl. Phys.* **90**, 1442-1454, doi:10.1063/1.1375019 %J Journal of Applied Physics (2001).
- 3 Lin, C. H., Friddle, P. A., Ma, C. H., Daga, A. & Chen, H. Effects of thickness on the electrical properties of metalorganic chemical vapor deposited Pb(Zr,Ti)O₃ (25–100 nm) thin films on LaNiO₃ buffered Si. *J. Appl. Phys.* **90**, 1509-1515, doi:10.1063/1.1383262 %J Journal of Applied Physics (2001).
- 4 Huang, G.-F. & Berger, S. Combined effect of thickness and stress on ferroelectric behavior of thin BaTiO₃ films. *J. Appl. Phys.* **93**, 2855-2860, doi:10.1063/1.1540225 %J Journal of Applied Physics (2003).
- 5 Hao, X. F. *et al.* Structural and ferroelectric transitions in magnetic nickelate PbNiO₃. *New J. Phys.* **16**, 015030, doi:10.1088/1367-2630/16/1/015030 (2014).
- 6 Yuan, S. *et al.* Room-temperature ferroelectricity in MoTe₂ down to the atomic monolayer limit. *Nat. Commun.* **10**, 1-6, doi:10.1038/s41467-019-09669-x (2019).
- 7 Qi, J., Wang, H., Chen, X. & Qian, X. Two-dimensional multiferroic semiconductors with coexisting ferroelectricity and ferromagnetism. *Appl. Phys. Lett.* **113**, doi:10.1063/1.5038037 (2018).
- 8 Gao, L. *et al.* Intrinsically elastic polymer ferroelectric by precise slight cross-linking. *Science* **381**, 540-544, doi:10.1126/science.adh2509 (2023).
- 9 Jiang, J. *et al.* Flexible ferroelectric element based on van der Waals heteroepitaxy. *Sci. Adv.* **3**, e1700121, doi:10.1126/sciadv.1700121 (2017).
- 10 Fujisawa, H., Nakashima, S., Kaibara, K., Shimizu, M. & Niu, H. Size effects of epitaxial and polycrystalline Pb(Zr, Ti)O₃ thin films grown by metalorganic chemical vapor deposition. *Jpn. J. Appl. Phys.* **38**, 5392, doi:10.1143/JJAP.38.5392 (1999).
- 11 Liu, F. *et al.* Room-temperature ferroelectricity in CuInP₂S₆ ultrathin flakes. *Nat. Commun.* **7**, 12357-12357, doi:10.1038/ncomms12357 (2016).
- 12 Io, W. F. *et al.* Temperature- and thickness-dependence of robust out-of-plane ferroelectricity in CVD grown ultrathin van der Waals α -In₂Se₃ layers. *Nano Res.* **13**, 1897-1902, doi:10.1007/s12274-020-2640-0 (2020).
- 13 Nishino, R., Fujita, T. C., Kagawa, F. & Kawasaki, M. Evolution of ferroelectricity in ultrathin PbTiO₃ films as revealed by electric double layer gating. *Sci. Rep.* **10**, 10864, doi:10.1038/s41598-020-67580-8 (2020).
- 14 Chandra, P., Dawber, M., Littlewood, P. B. & Scott, J. F. Scaling of the coercive field with thickness in thin-film ferroelectrics. *Ferroelectrics* **313**, 7-13, doi:10.1080/00150190490891157 (2004).
- 15 Jo, J. Y., Kim, Y. S., Noh, T. W., Yoon, J.-G. & Song, T. K. Coercive fields in ultrathin BaTiO₃ capacitors. *Appl. Phys. Lett.* **89** (2006).
- 16 Dawber, M., Chandra, P., Littlewood, P. B. & Scott, J. F. Depolarization corrections to the coercive field in thin-film ferroelectrics. *J. Condens. Matter Phys.* **15**, L393, doi:10.1088/0953-8984/15/24/106 (2003).
- 17 Xiao, J. *et al.* Intrinsic Two-dimensional ferroelectricity with dipole locking. *Phys. Rev. Lett.* **120**, 227601, doi:10.1103/PhysRevLett.120.227601 (2018).

- 18 Nordlander, J. *et al.* The ultrathin limit of improper ferroelectricity. *Nat. Commun.* **10**, 5591, doi:10.1038/s41467-019-13474-x (2019).
- 19 Cajipea, V. *et al.* Copper ordering in lamellar CuMP₂S₆ (M= Cr, In): transition to an antiferroelectric or ferroelectric phase. *Ferroelectrics* **185**, 135-138 (1996).
- 20 Susner, M., Rao, R., Pelton, A., McLeod, M. & Maruyama, B. Temperature-dependent raman scattering and x-ray diffraction study of phase transitions in layered multiferroic CuCrP₂S₆. *Phys. Rev. Mater.* **4**, 104003 (2020).
- 21 Miao, P. *et al.* Ferroelectricity and self-polarization in ultrathin relaxor ferroelectric films. *Sci. Rep.* **6**, 19965, doi:10.1038/srep19965 (2016).
- 22 Chernikova, A. *et al.* Ultrathin Hf_{0.5}Zr_{0.5}O₂ ferroelectric films on Si. *ACS Appl. Mater. Interfaces* **8**, 7232-7237, doi:10.1021/acsami.5b11653 (2016).
- 23 Cui, C. *et al.* Intercorrelated in-plane and out-of-plane ferroelectricity in ultrathin two-dimensional layered semiconductor In₂Se₃. *Nano Lett.* **18**, 1253-1258, doi:10.1021/acs.nanolett.7b04852 (2018).
- 24 Xi, Z. *et al.* High-temperature tunneling electroresistance on metal/ferroelectric/semiconductor tunnel junctions. *Appl. Phys. Lett.* **111** (2017).
- 25 Wu, J. *et al.* High tunnelling electroresistance in a ferroelectric van der Waals heterojunction via giant barrier height modulation. *Nat. Electron.* **3**, 466-472, doi:10.1038/s41928-020-0441-9 (2020).

REVIEWER COMMENTS

Reviewer #1 (Remarks to the Author):

The manuscript has been significantly improved. There are still some minor errors. For example, the vertical lines and signs (X and K) are not in the right positions in Fig. S14(a). I suggest that the authors invite a DFT expert to review the manuscript before publication.

Reviewer #2 (Remarks to the Author):

I am satisfied with the revisions and thus recommend publishing this work at Nat Comm.

Reviewer #3 (Remarks to the Author):

The revised manuscript has shown significant improvement. The authors have meticulously addressed numerous comments provided by myself and other reviewers, offering thorough and appropriate responses through various probing techniques. Notably, their endeavors to assess the ferroelectric properties of bulk CuCrP_2S_6 via the PUND method and to analyze the Cu sites in the paraelectric phase have substantially mitigated any ambiguity surrounding the existence and origin of room-temperature ferroelectricity in this sample. I thus believe that this work merits publication in Nature Communications.

REVIEWER COMMENTS

Reviewer #1 (Remarks to the Author):

The manuscript has been significantly improved. There are still some minor errors. For example, the vertical lines and signs (X and K) are not in the right positions in Fig. S14(a). I suggest that the authors invite a DFT expert to review the manuscript before publication.

Response: We thank for the precious comments from the reviewer. The vertical lines and X and K labels in Figure S14(a) have been carefully reviewed and corrected to the right positions. The revised band structure of Figure S14(a) was updated in the Supporting Information. Following the suggestion from the reviewer, we have invited a professional DFT theorist to review the simulation results in the manuscript and carefully proofread again to avoid similar errors.

Reviewer #2 (Remarks to the Author):

I am satisfied with the revisions and thus recommend publishing this work at Nat Comm.

Response: Thanks the reviewer for the encouraging comments. We have addressed all the comments and appended in the revised paper according to other reviewer's comments.

Reviewer #3 (Remarks to the Author):

The revised manuscript has shown significant improvement. The authors have meticulously addressed numerous comments provided by myself and other reviewers, offering thorough and appropriate responses through various probing techniques. Notably, their endeavors to assess the ferroelectric properties of bulk CuCrP_2S_6 via the PUND method and to analyze the Cu sites in the paraelectric phase have substantially mitigated any ambiguity surrounding the existence and origin of room-temperature ferroelectricity in this sample. I thus believe that this work merits publication in Nature Communications.

Response: Thanks the reviewer for the encouraging comments. We have addressed all the comments and appended in the revised paper according to other reviewer's comments.

REVIEWERS' COMMENTS

Reviewer #1 (Remarks to the Author):

I am satisfied with the current revision and I would like to recommend its publication at Nat Comm

REVIEWER COMMENTS

Reviewer #1 (Remarks to the Author):

I am satisfied with the current revision and I would like to recommend its publication at *Nat Comm*

Response: Thanks the reviewer for the encouraging comments and recommendation for publication in *Nat. Comm.*